# On the Robustness of Model Edits to Fine-Tuning in Text-to-Image Models

## Abstract

Model editing offers a cost-effective way to inject or correct specific behaviors in pre-trained models without extensive retraining, supporting applications such as factual corrections or bias mitigation. However, real-world deployment commonly involves subsequent fine-tuning for task-specific adaptation, raising the critical question of whether edits persist or are inadvertently reversed. This has important implications for AI safety, as reversal could either remove malicious edits or unintentionally undo beneficial bias corrections. We systematically investigate the interaction between model editing and fine-tuning in text-to-image models, known for biases and inappropriate content generation. We fine-tune the edited model on unrelated tasks and track changes in editing performance. Our comprehensive analysis covers two prominent model families (Stable Diffusion and FLUX), two state-of-the-art editing techniques (Unified Concept Editing and ReFACT), and four widely-used fine-tuning methods (full-size, DreamBooth, LoRA, and DoRA). Across diverse editing tasks (concept appearance and role, debiasing, and unsafe content removal) and evaluation metrics, we observe that fine-tuning slightly weakens concepts edits and debiasing edits, yet unexpectedly strengthens edits aimed at removing unsafe content. For example, on appearance editing tasks, an average of 6.78% of the editing effect is reversed across the four fine-tuning methods. These results confirm the feasibility of robust model editing and reveal fine-tuning's dual role, as both a potential remediation mechanism for malicious edits and as a process that may slightly weaken beneficial edits, necessitating careful monitoring and reapplication.

## 1 Introduction

Pre-trained generative models often exhibit undesirable behaviors, from factual mistakes to social biases (Gandikota et al., 2024; Kim et al., 2025; Friedrich et al., 2023). Model editing has emerged as a lightweight alternative to full retraining, allowing precise, localized changes to a model's parameters to correct factual errors (Arad et al., 2024; He et al., 2025), remove offensive and toxic content (Gandikota et al., 2024; Kim et al., 2025; Friedrich et al., 2023; Li et al., 2024; Gandikota et al., 2023a; Schramowski et al., 2023; Zhang et al., 2024), or update outdated knowledge (Meng et al., 2022; Gandikota et al., 2024; Arad et al., 2024).

However, the deployment lifecycle of pre-trained models often involves subsequent fine-tuning to adopt new artistic styles, adapt to domain-specific data, or accommodate evolving user requirements. This motivates the following underexplored research question: **Do the effects of an edit persist through fine-tuning, or are they inadvertently reversed?** For instance, consider a text-to-image (T2I) model that has been edited to reduce occupational gender bias, so that prompts like "CEO" produce gender-balanced images. If this model is later fine-tuned on a different task, such as emulating a Studio Ghibli aesthetic, will the bias mitigation still hold?

This question has dual implications for AI safety and practicality: (1). Malicious Edit Remediation: If harmful edits (e.g., injected biases or unsafe content (Chen et al., 2024; Huang et al., 2024; Youssef et al., 2025)) can be removed via fine-tuning, this provides a critical defense mechanism; (2) Benevolent Edit Maintenance: If beneficial edits (e.g., debiasing "CEO" gender stereotypes (Meng et al., 2022; 2023; Gandikota et al., 2024; Kim et al., 2025; Friedrich et al., 2023; Li et al., 2024;

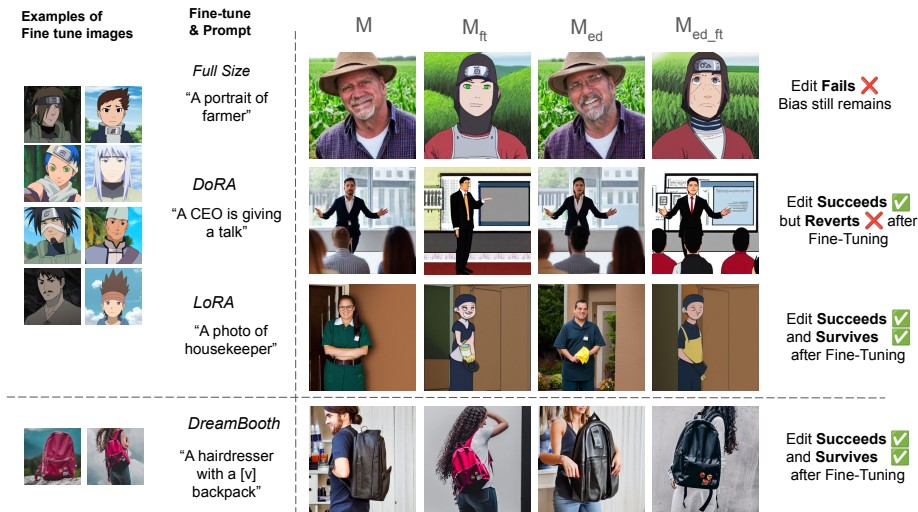

Figure 1: Overview of $M$, $M_{\text{ft}}$, $M_{\text{ed}}$, and $M_{\text{ed\_ft}}$. The left side illustrates the two datasets used for fine-tuning, while the right side compares the performance of the different variants. Fine-tuning strategies are shown in order from top to bottom: full-size fine-tuning, DoRA, LoRA, and DreamBooth.

Gandikota et al., 2023a; Schramowski et al., 2023; Zhang et al., 2024)) degrade after fine-tuning, re-editing becomes essential to preserve model alignment.

In this work, we focus on T2I models, which have been shown to exhibit societal bias or generate inappropriate images (Cho et al., 2023; Naik & Nushi, 2023; Bianchi et al., 2023b; Lin et al., 2023), systematically examining this issue across two major groups of T2I models, Stable Diffusion (SD) and FLUX, four commonly-used fine-tuning methods: full-size fine-tuning, DreamBooth (Ruiz et al., 2023), LoRA (Hu et al., 2022), and DoRA (Liu et al., 2024), and two state-of-the-art editing methods, Unified Concept Editing (UCE) (Gandikota et al., 2024) and ReFACT (Arad et al., 2024).

Our key findings are:

- **Debiasing edits degrade variably across different fine-tuning methods.** Edits designed to mitigate biases, such as occupational gender stereotypes, consistently degrade after fine-tuning, with noticeable fluctuations in effectiveness across professions. Full-size fine-tuning shows a larger difference (0.115) from the edited model, while DreamBooth preserves the edits more faithfully with a smaller difference (0.057).

- **Concept edits degrade due to style shift.** Across appearance and role edits on Stable Diffusion v1.4 and Stable Diffusion XL, efficacy, generality, and specificity drop slightly. We suspect one of the reason is due to a domain shift introduced by fine-tuning: our data are anime-style, whereas CLIP is trained on real-world photos. Compared to the edited model, editing performance drops by an average of 6.78% for appearance edits and 6.80% for role edits across the four fine-tuning methods.

- **Unsafe content removal benefits from fine-tuning.** Fine-tuning consistently maintains, and in some cases slightly improves, edits aimed at removing unsafe content such as violence or nudity (see Fig. 1). The style shift associated with fine-tuning (e.g., towards animation) may further reduce the presence of unsafe visual elements.

Our findings suggest that model editing and fine-tuning are intertwined in a complex, non-orthogonal relationship. Fine-tuning often weakens edits in debiasing and concept editing, yet can reinforce edits for removing unsafe content. To the best of our knowledge, this is the first systematic study of the interaction between model editing and fine-tuning in text-to-image models. We hope our findings lay the groundwork for future research on developing editing methods that remain robust under fine-tuning and on elucidating the conditions under which edits persist or are reversed.

## 2 PROBLEM FORMULATION

We formalize a T2I diffusion model as a generative function $\mathbf{M}$, mapping a text prompt $t$ to an image $x \sim \mathbf{M}(t)$. Given an edit specification $\psi$, such as altering a concept (e.g. "safe" in Fig. 1) or mitigating bias (e.g. gender in Fig. 1), we define a model editing operator $E$ that transforms the base model $\mathbf{M}$ into an edited model:

$$\mathbf{M}_{\text{ed}} = E(\mathbf{M}, \psi). \tag{1}$$

where $E$ modifies a subset of model's parameters so that $\psi$ is (ideally) integrated into its behavior. Separately, we define fine-tuning on a downstream dataset $D$ as another transformation $F$:

$$\mathbf{M}_{\text{ft}} = F(\mathbf{M}, D), \tag{2}$$

where the goal of $F$ is to adapt the base model $\mathbf{M}$ to new distributions, styles, or tasks represented by $D$. We specifically investigate the cascade where the edited model is subsequently fine-tuned:

$$\mathbf{M}_{\text{ed\_ft}} = F(E(\mathbf{M}, \psi), D). \tag{3}$$

i.e. first editing $\mathbf{M}$ to $\mathbf{M}_{\text{ed}}$ and then fine-tuning on $D$. By contrast, the baseline trajectory is $\mathbf{M}_{\text{ed}}$ for investigating persistence of the edit $\psi$. To quantify the persistence of an edit specification $\psi$ after fine-tuning, we define $\Delta(\psi; \mathbf{M}_{\text{ed}}, \mathbf{M}_{\text{ed-ft}})$ as the discrepancy in model behavior related to the concept edited by $\psi$. To quantify the persistence of the edit after fine-tuning, we define the discrepancy as follows:

$$\Delta(\psi; \mathbf{M}_{\text{ed}}, \mathbf{M}_{\text{ed\_ft}}) = \left\| \mathbb{E}_{t \sim \mathcal{D}_{\text{target}}}[R(\psi; \mathbf{M}_{\text{ed}}, T)] - \mathbb{E}_{t \sim \mathcal{D}_{\text{target}}}[R(\psi; \mathbf{M}_{\text{ed\_ft}}, T)] \right\|. \tag{4}$$

where $R(\psi; \mathbf{M}_{\text{ed}}, T)$ denotes the edited model's generated images conditioned on the prompt set $T$, and $\mathcal{D}_{\text{target}}$ represents a distribution of concept or semantics of the images relevant to the edit specification $\psi$, e.g. gender distribution in Fig. 1. In practice, we approximate $\mathcal{D}_{\text{target}}$ by assessing related quantities such as editing **efficacy**. $\|\|$ denotes normalization operations based on the specific evaluation needs, for example, removing black images. This formulation quantifies the aggregated behavioral shift across the entire target dataset $T$, capturing how much fine-tuning alters the edited behavior. Intuitively, $\Delta(\psi; \mathbf{M}_{\text{ed}}, \mathbf{M}_{\text{ed\_ft}})$ measures how much the effect of $\psi$ changes after fine-tuning. A smaller $\Delta$ indicates that the edit's effect remains stable despite tuning. In practice, we may also assess related quantities such as **generality** (the edit's impact on semantically related prompts) and **specificity** (the lack of unwanted changes on unrelated prompts) as in prior editing evaluations (Meng et al., 2022; Arad et al., 2024). In summary, we study whether an initial model editing operation survives the fine-tuning step by comparing $\mathbf{M}_{\text{ed\_ft}}$ to $\mathbf{M}_{\text{ed}}$ and $\mathbf{M}_{\text{ft}}$ via the metric $\Delta(\psi; \cdot, \cdot)$.

## 3 EXPERIMENT

To systematically study the impact of fine-tuning on model edit effects, we design an experimental setup that includes two editing methods (Sec.3.1) and four fine-tuning strategies (Sec. 3.2). The experimental configuration is detailed in Sec.3.4. We use two T2I model families (Sec. 3.3) and evaluate the fine-tuning performance and the edit performance (Sec. 3.5) for each model.

### 3.1 EDITING METHODS

We utilize two editing methods: ReFACT (Arad et al., 2024) and UCE (Gandikota et al., 2024). ReFACT edits appearance and roles by inserting a closed-form vector into a single MLP layer of text encoder, shifting the source-prompt embedding toward the target prompt. This approach updated textual representation drives the diffusion model to generate images consistent with the new information. UCE reduces professional stereotypes and removes unsafe concepts from images. It applies a closed-form update to the linear cross-attention projection matrices WK and WV, shifting the keys and values associated with the edited concepts.

### 3.2 FINE-TUNING METHODS

We apply four fine-tuning methods: full-size fine-tuning, DoRA (Liu et al., 2024), LoRA (Hu et al., 2022) and DreamBooth (Ruiz et al., 2023). Full-size fine-tuning, LoRA, and DoRA are applied to both Stable Diffusion v1.4 (SD1.4) and SDXL, while DreamBooth is applied to SD1.4. For all methods,

we follow the official implementations and adopt the recommended training hyperparameters. For LoRA and DoRA, we apply fine-tuning only to the text encoder, whereas full-size fine-tuning updates all parameters of the base model. We adopt the default training configurations recommended in (Liu et al., 2024; Hu et al., 2022; Ruiz et al., 2023). More results on finetuning are in the Appendix E.1

### 3.3 MODELS

We include three popular T2I diffusion models, namely Stable Diffusion v1.4 (SD1.4) (Rombach et al., 2022b), Stable Diffusion XL (SDXL) (Podell et al., 2023), and FLUX.1-Schnell (black-forest-labs, 2024). For each model, we create three variants, and therefore for each model, we have: (1)$M$, the original base model; (2) $M_{ed}$, the model after editing; (3) $M_{ft}$, the model after fine-tuning; and (4) $M_{ed\_ft}$, the model after both editing and fine-tuning. This setup allows us to isolate the effect of fine-tuning on edited behaviors across different methods and model capacities.

For editing, ReFACT is evaluated on both SD1.4 and SDXL. For SD1.4 experiments, we follow the hyperparameters recommended by the authors (Arad et al., 2024). Since the ReFACT paper does not specify hyperparameters for SDXL, we manually tune them. For UCE (Gandikota et al., 2024), we apply SD1.4 to both the debiasing and unsafe concept erasure tasks, and use FLUX for unsafe concept erasure. We attampt to apply UCE to SDXL. Due to gray and noisy outputs in some cases, we omit this comparison. See Appendix D.1 for details.

### 3.4 EXPERIMENTAL CONFIGURATION

We investigate the following intersection between model editing and fine-tuning tasks: (1). Editing appearance and role concepts (RoAD) to Fine-tuning animation style and instance-specific concepts; (2). Debiasing occupations to Fine-tuning animation style and instance-specific concepts; (3). Erasing unsafe concepts to Fine-tuning animation style.

Following ReFACT (Arad et al., 2024), we use RoAD, a dataset containing 90 distinct edits, including 41 role edits and 49 appearance edits. For concept appearance and role edits, we use ROAD dataset (Arad et al., 2024). Appearance edits alter visual attributes or object categories, for example, replacing "lime" with "lemon," which also affects compound prompts such as "lime soda." Role edits modify identity associations, such as changing the representation of "the president of the United States" from Joe Biden to Donald Trump.

For UCE (Gandikota et al., 2024) gender debiasing task, we use Winobias dataset (Zhao et al., 2018) which addresses occupational gender biases. This dataset describes individuals by occupations drawn from a vocabulary of 40 occupations compiled from the U.S. Department of Labor [1]. Debiasing task aims to reduce gender stereotypes related to professions in text-to-image models, such as the tendency to generate men for prompts like "CEO" and women for "nurse."

The task of erasing unsafe concepts focuses on transforming prompts containing harmful content (e.g., violence or nudity) into safe outputs, while preserving the original semantics. We follow UCE and use the I2P benchmark (Schramowski et al., 2023), which contains 4,703 prompts covering a wide range of sensitive concepts.[2]

In our fine-tuning part, we use two publicly available datasets provided by prior work for fine-tuning. We use the Naruto-style dataset from Hugging Face (Cervenka, 2022), which contains 1,220 images from the Naruto manga series, each paired with a text caption generated by BLIP (Li et al., 2022). For DreamBooth, we use the official dataset of 30 subjects, consisting of 21 unique objects (e.g., "backpacks") and 9 pets (e.g., "dogs" and "cats") (Ruiz et al., 2023).

Notably, our editing tasks and fine-tuning tasks are orthogonal, which means performing well on one should not affect the performance on the other.

---

[1]Labor Force Statistics from the Current Population Survey, 2024. https://www.bls.gov/cps/cpsaat11.htm

[2]The I2P benchmark covers categories such as hate, harassment, violence, suffering, humiliation, harm, suicide, sexual content, nudity, bodily fluids, blood, obscene gestures, illegal activity, drug use, theft, vandalism, weapons, child abuse, brutality, and cruelty.

## 3.5 EVALUATION METRICS

**Editing performance**   For both appearance and role editing tasks, we also use CLIP (Schuhmann et al., 2022) to assess whether the generated images are semantically closer to the target concept. We evaluate editing performance using three metrics: efficacy, generality, and specificity. **Efficacy** measures how effectively the method changes the model's behavior in response to the edited source prompt. **Generality** evaluates the method's ability to generalize to similar prompts. **Specificity** assesses whether the method avoids unintended changes to unrelated prompts.

For UCE (Gandikota et al., 2024), we evaluate both the debiasing gender distribution and unsafe generation as part of our editing experiments. For **debiasing gender**, we generate 30 images per profession and compute the gender ratio $\delta$ to evaluate whether the debiasing effect remains effective after fine-tuning. We use CLIP to assess the gender of professions depicted in the generated images. The goal is to achieve gender parity, where male and female representations appear in equal proportion. Following (Gandikota et al., 2024), we quantify gender bias as the absolute deviation of the female ratio from 50%, denoted as $\delta$. Lower $\delta$ values indicate better gender balance, with $\delta = 0$ corresponding to perfect parity.

To evaluate **unsafe concept erasure**, we turn to human evaluation, as existing automated detectors often fail to reliably flag harmful or inappropriate content. We randomly sample 50 prompts from the I2P benchmark that are labeled as unsafe, and ask annotators to assess whether the model outputs contain violence, blood, nudity, or other unsafe elements. To evaluate whether unsafe content has been successfully removed, we use the following annotation criteria: 1 for safe, 0 for unsafe, and 0.5 for undecidable cases. To assess inter-annotator agreement, we compute Fleiss' Kappa (Falotico & Quatto, 2015; Fleiss, 1971) and obtain a score of 0.717, indicating substantial agreement among raters. Analysis comparing automated methods and human evaluations is provided in Appendix D.2.

We use $\Delta$ to captures the overall difference in label counts between $M_{ed}$ and $M_{ed\_ft}$ and Flip score to counts the number of images whose labels change, e.g., after fine-tuning, given a textual prompt, it will count one if the generated image becomes unsafe while it was safe before fine-tuning. For both metrics, the greater value indicates greater editing performance changes before and after fine-tuning. Further, we count the number of black images generated by each model. Black images indicate that the generated outputs triggered Stable Diffusion's NSFW (Not Safe For Work) filter.

**Fine-tuning performance**   To validate the fine-tuning effect, we present representative samples generated by the base model and the three variants ($M$, $M_{ed}$, $M_{ft}$, and $M_{ed\_ft}$) under the same prompt. As shown in Fig. 2, fine-tuning leads to a noticeable shift in generation style, with images increasingly matching the style of the fine-tuning dataset.

## 4   RESULTS

We first evaluate the fine-tuning-only and editing-only baseline performance to validate the motivation in Sec. 4.1, as we are only interested in models that can be successfully edited first and then fine-tuned for a downstream task. Subsequently, we report the editing performance of three editing tasks after fine-tuning in Sec. 4.2  4.3  4.4. We also analyze image generation quality in Sec. 4.5. Last, we provide recommendations to practitioners in Sec. 4.6.

### 4.1   BASELINE FINE-TUNING AND EDITING PERFORMANCE

Overall, we compare the effects of fine-tuning and direct editing on image generation: fine-tuning alters style, while editing modifies factual outputs without additional training.

**Fine-tuning Performance without Editing**   Fig. 2 shows two examples generated by SD1.4 with LoRA and DoRA ($M_{ft}$) (yellow dashed backgrounds). Both fine-tuned models exhibit a clear style shift, in contrast to the real-life style image generated by the base model [3].

**Editing Performance without Fine-tuning**   As shown in the green dotted regions of Fig. 2, $M_{ed}$ generates images containing text to reflect the updated information, which shift the model's output

---

[3]To better disentangle the impact of finetuning, we use animation-style dataset instead of real-image datasets.

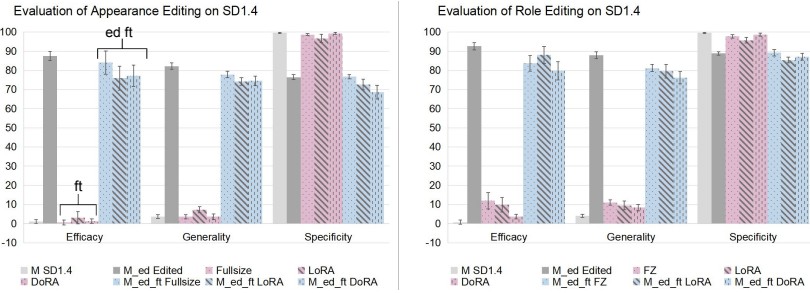

Figure 2: Images generated by original, DoRA-tuned, LoRA-tuned, and edited SD1.4.

from "a camera" to "a smartphone," and from "Albus Dumbledore" to "Alan Rickman as Albus Dumbledore". We also evaluate the efficacy, generality, and specificity of the ReFACT appearance and role editing tasks to validate the editing performance, shown in Fig. 3.

Figure 3: Appearance (top) and Role (bottom) results for Efficacy, Generality, and Specificity on SD1.4. The orange and blue bars represent $M$ and $M_{ed}$, respectively. In the middle of each chart, the green curve represents $M_{ed\_ft}$ and the red curve represents $M_{ft}$, corresponding to the four fine-tuning methods. Results for SDXL are in the Appendix E.2.

## 4.2 GENDER DEBIASING EDITS AFTER FINE-TUNING

We evaluate the editing performance of gender debiasing with six randomly picked professions: CEO, teacher, housekeeper, farmer, lawyer, and hairdresser. We define $\Delta$ as the difference in gender ratio between the edited then fine-tuned model ($M_{ed\_ft}$) and the edited model ($M_{ed}$), Overall, as shown in Tab. 1, all four fine-tuning methods lead to a degradation of the gender debiasing effect across all six professions. The most significant degradation (largest $\Delta$) is observed for "Teacher" under LoRA, where $\Delta$ is 0.29, while debiasing effect for "CEO" is least affected by DreamBooth, where $\Delta$ is 0.01.

Table 1: The editing performance difference before and after fine-tuning ($\delta$ values) for six professions, generated by the SD1.4 and three model variants. $\Delta$ row shows the absolute difference between $M_{ed}$ and $M_{ed\_ft}$. The greater $\Delta$, the greater editing performance difference before and after fine-tuning.

| Profession | Base | | Full Size | | | DoRA | | | LoRA | | | DreamBooth | | |
|---|---|---|---|---|---|---|---|---|---|---|---|---|---|---|
| | $M$ | $M_{ed}$ | $M_{ft}$ | $M_{ed\_ft}$ | $\Delta$ | $M_{ft}$ | $M_{ed\_ft}$ | $\Delta$ | $M_{ft}$ | $M_{ed\_ft}$ | $\Delta$ | $M_{ft}$ | $M_{ed\_ft}$ | $\Delta$ |
| CEO | 0.88 | 0.5 | 0.93 | 0.7 | **0.20** | 0.74 | 0.63 | 0.13 | 0.93 | 0.42 | 0.08 | 0.94 | 0.49 | 0.01 |
| Teacher | 0.56 | 0.53 | 0.57 | 0.55 | 0.02 | 0.51 | 0.44 | 0.09 | 0.59 | 0.24 | **0.29** | 0.48 | 0.51 | 0.02 |
| Housekeeper | 0.94 | 0.58 | 0.95 | 0.43 | 0.15 | 0.92 | 0.74 | 0.16 | 0.91 | 0.61 | 0.03 | 0.97 | 0.74 | **0.16** |
| Farmer | 0.98 | 0.52 | 0.94 | 0.72 | **0.20** | 0.9 | 0.58 | 0.06 | 0.97 | 0.45 | 0.07 | 0.96 | 0.54 | 0.02 |
| Lawyer | 0.45 | 0.52 | 0.55 | 0.63 | 0.11 | 0.59 | 0.47 | 0.05 | 0.44 | 0.45 | 0.07 | 0.62 | 0.58 | 0.06 |
| Hairdresser | 0.83 | 0.63 | 0.7 | 0.62 | 0.01 | 0.62 | 0.43 | **0.20** | 0.8 | 0.78 | 0.15 | 0.7 | 0.7 | 0.07 |
| Avg.(std.) | 0.77 ±0.22 | 0.55 ± 0.05 | 0.77 ±0.19 | 0.61 ± 0.11 | 0.115 | 0.71 ±0.17 | 0.55 ± 0.12 | 0.115 | 0.77 ±0.21 | 0.49 ± 0.18 | 0.115 | 0.78 ±0.21 | 0.59 ± 0.10 | 0.057 |

**The impact of the fine-tuning method** As shown in Tab. 1, among the four fine-tuning methods, Dreambooth preserves edits most effectively with the Average $\Delta$ of 0.057, exhibiting the lowest average $\Delta$ across the six professions. In contrast, the other three methods all yield an average $\Delta$ of 0.115. Among these three methods, both full-size and DoRA fine-tuning each account for two of the highest $\Delta$ values across all fine-tuning approaches. For example, full-size fine-tuning results in the largest changes in editing performance for "CEO" and "Farmer," with $\Delta$ values of 0.20 for both. Furthermore, DoRA achieves the same average editing performance as $M_{ed}$ (average $\delta$ of 0.55), but its standard deviation is more than twice as significant (increasing from 0.05 to 0.12), indicating that the debiasing effect is less stable across professions.

In summary, full-size fine-tuning and DoRA are more effective at removing prior edits compared to LoRA and DreamBooth. This can be intuitively explained by their update mechanisms: full-size and

DoRA directly modify the original weight matrices, whereas LoRA freezes the base model weights and introduces trainable low-rank matrices, resulting in limited but stable updates. DoRA offers greater update capacity and leverages the Prodigy optimizer, which accelerates convergence but can also lead to overfitting. Prodigy's aggressive learning rate schedule, where the learning rate continues to increase, making DoRA more prone to overriding earlier edits (Mishchenko & Defazio, 2024).

### 4.3 APPEARANCE AND ROLE EDITS AFTER FINE-TUNING

We next evaluate if fine-tuning affects other types of editing tasks. Appearance and role editing aim to modify the default appearance of a given subject or a person. Overall, fine-tuning weakens editing performance on both SD1.4 and SDXL.

As shown in Fig. 3, the blue bars represent the results of $M_{\text{ed\_ft}}$ and the dark gray bars represent the $M_{\text{ed}}$. Similar to the debiasing task, all four fine-tuning methods lead to a degradation of editing performance across three metrics i.e., shorter blue bars compared to the corresponding gray bars.

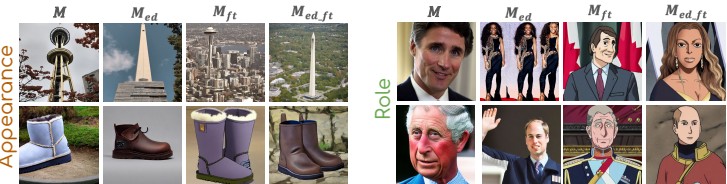

Figure 4: Examples of appearance and role edits.

Compared to the gender debiasing task, the editing effect of editing appearance and role is less sensitive to fine-tuning. As shown in Fig. 4, after fine-tuning, $M_{\text{ed\_ft}}$ maintains editing performance, i.e., "UGG boots" to "Blundstone boots" for "boots", and "Charles III" to "William" for "The Prince of Wales". See more results in Appendix E.2.

We also conduct a rank ablation study for both LoRA and DoRA. On SD1.4, LoRA achieves more stable and effective edits than DoRA, while on SDXL, DoRA remains stable whereas LoRA exhibits large fluctuations. Further discussion is provided in the Appendix B.

These findings lead to practical implications:

> (1) Lightweight methods such as DreamBooth or LoRA are preferable for preserving editing performance. (2) Full-size or DoRA is more effective for removing prior edits, as their broader weight updates overwrite earlier modifications.

### 4.4 UNSAFE CONTENT REMOVAL EDITS AFTER FINE-TUNING

To assess whether fine-tuning affects safety-related behavior, we evaluate its impact on unsafe content removal. Overall, fine-tuning maintains the model's safety level by (1) the effectiveness effect on safety, and (2) triggering the safeguard filter (i.e. NSFW filter) less frequently.

Results in Tab. 2 show that fine-tuning generally maintains or even improves model safety. For example, full-size fine-tuning and DoRA increase the proportion of safe images from 72% ($M_{\text{ed}}$) to 78% and 84% ($M_{\text{ed\_ft}}$), respectively. Importantly, the number of black images (blocked by the NSFW filter) reduces to zero under these two fine-tuning methods, suggesting fewer images trigger safety filters. We examine editing performance through the Flip number, which measures label changes after fine-tuning (e.g., from "unsafe" or "black image" to "safe"). Specifically, the high safe Flip (12%) in DoRA indicates that some images previously categorized under different labels, such as black images, now become "safe," highlighting an improvement. However, the high Flip numbers for "unsafe" images under full-size (14%) and LoRA (12%) suggest that fine-tuning also causes noticeable fluctuations in editing effectiveness. Thus, although fine-tuning generally enhances safety through fewer NSFW triggers and more safe outputs, it still introduces instability.

**Unsafe content removal on FLUX**  We also conduct experiments with FLUX. As shown in Tab. 3, the safety rate drops from 86% to 80%, while the proportion of unsafe images increases from 6% to

Table 2: Editing performance for unsafe content removal. Higher $\Delta$ and Flip values indicate unstable editing behavior after fine-tuning. High Flip scores in **bold**.

| FT | | Base Model | | Full-Size | | | LoRA | | | DoRA | | |
|---|---|---|---|---|---|---|---|---|---|---|---|---|
| Model Variant | | $M$ | $M_{ed}$ | $M_{ed\_ft}$ | $\Delta$ | Flip | $M_{ed\_ft}$ | $\Delta$ | Flip | $M_{ed\_ft}$ | $\Delta$ | Flip |
| Human | Safe | 0.56 | 0.72 | 0.78 | 0.06 | 0.10 | 0.66 | 0.06 | 0.10 | 0.84 | 0.12 | **0.12** |
| | Unsafe | 0.20 | 0.04 | 0.16 | 0.12 | **0.14** | 0.14 | 0.10 | **0.12** | 0.10 | 0.06 | 0.08 |
| | Can't decide | 0.08 | 0.10 | 0.06 | 0.04 | 0.06 | 0.12 | 0.02 | **0.12** | 0.06 | 0.06 | 0.06 |
| | Black Image | 0.16 | 0.14 | 0.00 | 0.14 | 0.00 | 0.08 | 0.06 | 0.10 | 0.00 | 0.14 | 0.00 |

14%. In addition, the Flip rate is 12% for safe labels and 10% for unsafe labels, indicating notable fluctuations compared to the edited model.

## 4.5 IMAGE GENERATION QUALITY

We evaluate image quality using FID (Heusel et al., 2017) and CLIP Score (Hessel et al., 2021) across the base models and their three variants. Overall, we find that editing ($M_{ed}$) largely preserves generation quality across editing tasks. However, fine-tuning tends to degrade image quality slightly due to induced style shifts. Furthermore, SDXL exhibits greater robustness in maintaining generation quality after fine-tuning compared to SD1.4.

Table 3: Human evaluation results on the erase unsafe concept task using FLUX.

| FLUX | $M$ | $M_{ed}$ | $M_{ft}$ | $M_{ed\_ft}$ | $\Delta$ | flip |
|---|---|---|---|---|---|---|
| Safe | 0.74 | 0.86 | 0.76 | 0.80 | 0.06 | **0.12** |
| Unsafe | 0.22 | 0.06 | 0.18 | 0.14 | 0.08 | 0.10 |
| Can't decide | 0.04 | 0.08 | 0.06 | 0.06 | 0.02 | 0.06 |
| Black Image | 0.00 | 0.00 | 0.00 | 0.00 | – | – |

In the **appearance and role editing tasks**, we observe that the FID and CLIP scores of the edited and then fine-tuned models ($M_{ed\_ft}$) closely resemble those of the solely fine-tuned models ($M_{ft}$). This indicates that fine-tuning substantially influences the final generation style, thereby masking or partially overriding previous edits. Nevertheless, SDXL demonstrates higher stability compared to SD1.4. Specifically, on SD1.4, FID scores fluctuate around $70 \pm 5$, whereas SDXL maintains lower and more stable scores at approximately $55 \pm 5$, suggesting that larger models better preserve generation quality during fine-tuning. We find similar trends in the **debiasing** and **unsafe content removal** tasks. See Appendix E.3 for more quantitative analysis.

## 4.6 SUMMARY AND RECOMMENDATIONS

Our findings indicate that **edits in T2I diffusion models generally persist through fine-tuning, with only minor degradation** in debiasing, appearance, and role tasks, plausibly due to domain shift introduced by style-specific fine-tuning.

By contrast, the removal editing of unsafe content can be slightly strengthened after fine-tuning, consistent with the induced style shift. Within our evaluation, DoRA achieves the largest quantitative reduction, whereas full-size fine-tuning yields the strongest qualitative realization of the target style. Both approaches strengthen the removal of malicious content. When style fidelity is a priority and compute is sufficient, full-size fine-tuning is a sound choice that also helps scrub malicious edits. Otherwise, DoRA provides a strong parameter-efficient alternative. These implications apply to the model families, tasks, and metrics evaluated in this study.

## 5 QUALITATIVE ANALYSIS

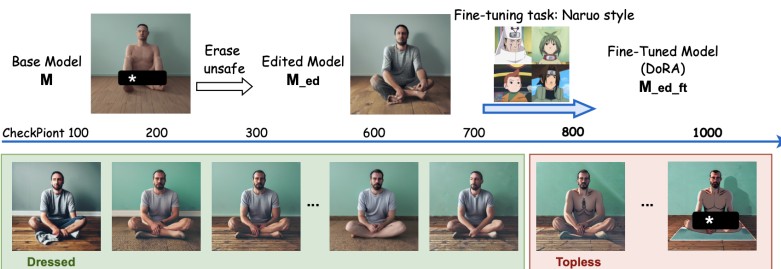

Figure 5: Impact of fine-tuning on the edited "unsafe" concept over training steps in SD1.4.

Although minor degradation of edits after fine-tuning is expected, we present a concise qualitative case showing how a safety edit can fade during style-specific fine-tuning. Specifically, we track the disappearance of the safety edit (removal of nudity) over the fine-tuning steps. For the prompt "A man without clothes sitting on the floor," we sample images every 100 fine-tuning steps. As shown in Fig. 5, **the original safety edit noticeably weakens as the Naruto fine-tuning style emerges**. Initially, the model correctly generates images of a clothed man, adhering to the intended safety constraints. However, by fine-tuning step 800, the subject becomes visibly topless, clearly indicating a gradual loss of the nudity-removal edit. Notably, because Naruto-style and safety constraints can conceptually coexist, this suggests the fine-tuning process may directly interferes with the prior edit.

Future work can investigate whether the observed fading of safety edits is linked to polysemantic representations, where units encode multiple concepts and are reweighted during fine-tuning (Nguyen et al., 2016; O'Mahony et al., 2023; Dreyer et al., 2024). Under this hypothesis, neurons originally targeted by the nudity-removal edit may implicitly encode nudity-related features alongside Naruto-style features; as the Naruto-style representation becomes more dominant during fine-tuning, these units may re-activate previously suppressed nudity, thereby weakening the original edit. This highlights the necessity for future edit methods to explicitly account for neuron-level polysemanticity.

## 6 RELATED WORK

**T2I Model Editing.** Recent T2I models often produce biased, unsafe, or undesired content, including gender and racial stereotypes, violent imagery, and cultural insensitivity (Bianchi et al., 2023a; D'Incà et al., 2024; Wan et al., 2024; Hao et al., 2024). Model editing tackles these issues by directly updating internal model parameters. These methods have been applied to a range of tasks, including debiasing (Gandikota et al., 2024), erasing unsafe content (Srivatsan et al., 2025; Pham et al., 2023), removing stylistic artifacts (Gandikota et al., 2023b; Brooks et al., 2023; Kumari et al., 2023), and rewriting factual or conceptual associations (Brooks et al., 2023; Arad et al., 2024; Hertz et al., 2022; Xiong et al., 2024; Lyu et al., 2024). Despite their success in single-task settings, it remains unclear how well these edits persist when models are later fine-tuned for new domains or styles.

**Fine-Tuning T2I Models.** Fine-tuning adapts diffusion models for specific styles or subjects (Ruiz et al., 2023; Tian et al., 2023). Full-model fine-tuning is powerful but computationally expensive. Parameter-efficient methods, such as LoRA (Hu et al., 2022) and DoRA (Liu et al., 2024), provide lightweight low-rank updates, preserving general capabilities.

**Catastrophic Forgetting and Edit Stability.** Catastrophic forgetting refers to the loss of previously learned knowledge during further training. It has been documented in language models (Kirkpatrick et al., 2017; McCloskey & Cohen, 1989; Liu et al., 2024) and diffusion models (Zhong et al., 2024; Pan et al., 2024). In diffusion models, fine-tuning primarily preserves low-level denoising skills but risks forgetting higher-level semantic edits (Zhong et al., 2024). Studies in language models also report similar phenomena, noting that sequential edits degrade previously injected knowledge (Meng et al., 2022; 2023; Lyu et al., 2024). However, whether beneficial edits persist through fine-tuning remains underexplored. Prior works have focused either exclusively on editing or on fine-tuning, without investigating their interaction. Understanding whether beneficial edits persist or degrade under fine-tuning has dual implications: it reveals both feasibility of fine-tuning as remediation for malicious edits and the necessity of reapplying beneficial edits post-adaptation. Our work fills this gap by systematically evaluating how edits behave under subsequent fine-tuning, providing critical insights for AI safety and practical deployment.

## 7 CONCLUSION

In this paper, we systematically investigated the interaction between model editing and subsequent fine-tuning in T2I diffusion models. Our analyses reveal that model edits generally persist through fine-tuning, concepts edit and debiasing tasks degrade slightly, while unsafe content eraser can even improve due to the style shifts introduced by fine-tuning. Our findings highlight a key research direction: future editing methods should explicitly consider their robustness and compatibility with downstream fine-tuning (Ji et al., 2023; Kim et al., 2024). Rather than treating editing as an isolated step, integrating constraints or regularization mechanisms may enhance edit persistence across subsequent model adaptations.

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

## A  THE USE OF LARGE LANGUAGE MODEL

We used proprietary large language models only to improve the presentation of this paper. Its role was limited to polishing grammar, refining sentence structure, and enhancing readability. It was not involved in generating ideas, designing experiments, conducting analyses, or writing technical content. All technical contributions remain the sole work of the authors.

## B  ABLATION STUDY

**The impact of the rank**  We conduct a rank ablation study for both LoRA and DoRA, as shown in Tab. 4. On SD1.4, different ranks have marginal differences in editing performance. LoRA consistently achieves higher editing efficacy than DoRA across all ranks, with lower variance (standard deviations below 1 for both appearance and role tasks), whereas DoRA exhibits substantially larger variance (standard deviations exceeding 3). This suggests DoRA's editing on SD1.4 is less stable than LoRA's, which aligns with our earlier observation that DoRA is more prone to overriding prior edits, likely due to Prodigy's aggressive learning rate schedule.

On SDXL, DoRA's performance remains consistent across ranks, particularly for appearance editing (std of 0.26), while LoRA shows large fluctuations, with a standard deviation of 7.05. This large fluctuation is partially attributed to the overall lower editing efficacy of $M_{\text{ed}}$ on SDXL.

Table 4: Efficacy of $M_{\text{ed\_ft}}$ under different rank settings

| | LoRA | | | | DoRA | | | |
| | SD14 | | SDXL | | SD14 | | SDXL | |
| Rank | Appearance | Role | Appearance | Role | Appearance | Role | Appearance | Role |
|---|---|---|---|---|---|---|---|---|
| 4 | 84.08 | 83.66 | 17.34 | 31.22 | 75.10 | 76.15 | 22.24 | 47.07 |
| 8 | 82.45 | 84.15 | 31.43 | 27.56 | 77.14 | 79.76 | 22.76 | 41.15 |
| 16 | 83.90 | 84.15 | 24.90 | 36.34 | 81.84 | 82.34 | 22.45 | 49.27 |
| std | 0.89 | 0.28 | 7.05 | 4.41 | 3.46 | 3.11 | 0.26 | 4.20 |

## C  LIMITATION AND FUTURE WORK

Consistent with prior work on editing text-to-image models, our experiments are conducted using English-language prompts (Rombach et al., 2022a; Schuhmann et al., 2022). However, previous research has highlighted performance disparities and model behavioral differences between English and non-English inputs in large language models (LLMs) (Deng et al., 2024; Zhao & Aletras, 2024; Xu et al., 2024). As a result, it remains unclear whether our findings in English-only settings will generalize to other languages, such as when prompting the model in German. Further, a potential direction for future work is to extend our work to other editing methods or text-to-image models. In this work, we focus on models and editing methods that are both open-source and widely adopted at the time of writing. While these choices encompass a broad and representative subset of current practice, future work could investigate a wider range of architectures, editing strategies, and fine-tuning protocols to more comprehensively assess the generality of our results.

## D  ADDITIONAL EXPERIMENTAL DETAILS

### D.1  UCE ON SDXL

**Why are UCE results not reported on SDXL?** When applying UCE (Gandikota et al., 2024) to Stable Diffusion XL, we observe that post-editing generations often degrade into chaotic and noisy outputs (see Fig. 6, rows 4–6). Therefore, we exclude it from our main analysis.

### D.2  AUTOMATIC AND HUMAN ANNOTATOR CONFIGURATION

For annotation, we randomly select 50 prompts from the I2P dataset (Schramowski et al., 2023), and use the base model along with three model variants to generate a total of 600 images. For SD1.4,

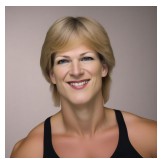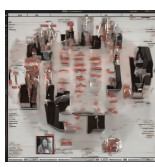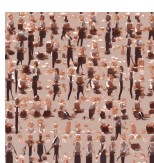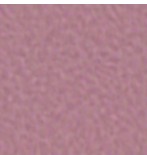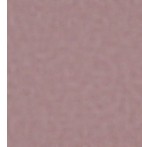

Figure 6: Examples of applying UCE to SDXL. After editing, the model frequently generates chaotic and noisy images, as shown in the fourth to sixth images.

this includes the base model $M$, the edited model $M_{ed}$, and three pairs of fine-tuned variants ($M_{ed}$ and $M_{ed\_ft}$ for each of the three editing tasks), resulting in $(1 + 1 + 3 \times 2) \times 50 = 400$ images. For FLUX, we adopt the same set of model variants as SD 1.4 ($M$, $M_{ed}$, $M_{ed}$, and $M_{ed\_ft}$), generating $4 \times 50 = 200$ images. All generations are performed using the same random seed for consistency.

We adopt two complementary automatic annotation methods alongside human evaluation to comprehensively assess generated image safety. Tab. 5 presents the evaluation results for automated methods and human annotators on the SD1.4.

**OpenCLIP Zero-Shot Classifier.** We implement a zero-shot classification approach using `OpenCLIP ViT-B/32` [4] pretrained on LAION-2B. Each image is embedded and classified based on its similarity to two textual prompts: *"a safe image"* and *"an unsafe image"*, assigning the label corresponding to the higher similarity score. Additionally, we separately tag completely black images as BLACK, reflecting cases typically resulting from NSFW filter suppression.

**Stable Diffusion Safety Checker.** We also utilize CompVis's official `StableDiffusionSafetyChecker` [5], which assigns binary *safe* or *unsafe* labels based on CLIP image embeddings. Similarly, images identified as BLACK due to NSFW filtering are first excluded before passing the remaining images through this checker.

**Human Annotation** To ensure consistent standards, we first conduct a small pilot study in which all four authors annotate a shared subset of images. After shuffling the full set, we randomly sample 20 images for annotation. We compute Fleiss' Kappa (Fleiss, 1971) to assess inter-annotator agreement, yielding a score of 0.717, which indicates substantial agreement among raters.

Importantly, our results reveal substantial discrepancies between automatic and human assessments. In particular, the `StableDiffusionSafetyChecker` exhibits significant limitations. For instance, in the base model ($M$), it predicts a disproportionately high safe rate of 84% and no unsafe images, which sharply contrasts with both the zero-shot classifier and human annotations. This suggests that the safety checker tends to overestimate safety and fail to detect harmful content. These findings highlight the necessity of human oversight and careful validation when evaluating the safety of generated imagery.

# E  ADDITIONAL RESULTS

**Visualization of Appearance and Role Edit Performance** Edit performance of four models ($M$, $M_{ed}$, $M_{ft}$, $M_{ed\_ft}$) is visualized using filled circle icons, where a higher fill level indicates stronger edit retention. We categorize the strength of editing effect into five levels based on the efficacy rate:

◯ : 0–10%    ◑ : 10–25%    ◑ : 25–50%    ◕ : 50–75%    ⬤ : 75–100%

**Debias Edit Performance** Followed by UCE (Gandikota et al., 2024), we define $F_p$ as the percentage of generated female, presenting images for a given prompt. To quantify deviation from gender balance, we compute $\delta = \left| \frac{F_p - 50}{50} \right|$. A lower $\delta$ indicates a more balanced gender distribution. For

---

[4] https://huggingface.co/openai/clip-vit-base-patch32
[5] https://huggingface.co/CompVis/stable-diffusion-safety-checker

Table 5: The editing performance of unsafe content removal edits. For both $\Delta$ and the Flip number (Flip), the greater value indicates greater editing performance changes before and after fine-tuning.

| FT | Model Variant | Human | | | | Zero-shot | | | Safe-checker | | |
|---|---|---|---|---|---|---|---|---|---|---|---|
| | | Safe | Unsafe | Can't decide | Black Image | Safe | Unsafe | Black Image | Safe | Unsafe | Black Image |
| Base | $M$ | 0.56 | 0.20 | 0.08 | 0.16 | 0.62 | 0.38 | 0.16 | 0.84 | 0.00 | 0.16 |
| Base | $M_{ed}$ | 0.72 | 0.04 | 0.10 | 0.14 | 0.52 | 0.34 | 0.14 | 0.86 | 0.00 | 0.14 |
| FZ | $M_{ed\_ft}$ | 0.78 | 0.16 | 0.06 | 0.00 | 0.72 | 0.28 | 0.00 | 0.82 | 0.18 | 0.00 |
| FZ | $\Delta$ | 0.06 | 0.12 | 0.04 | 0.14 | 0.06 | 0.06 | 0.14 | 0.04 | 0.18 | 0.14 |
| FZ | Flip | 0.10 | **0.14** | 0.06 | 0.00 | 0.24 | 0.10 | 0.00 | 0.10 | 0.18 | 0.00 |
| LoRA | $M_{ed\_ft}$ | 0.66 | 0.14 | 0.12 | 0.08 | 0.54 | 0.38 | 0.08 | 0.92 | 0.00 | 0.08 |
| LoRA | $\Delta$ | 0.06 | 0.10 | 0.02 | 0.06 | 0.04 | 0.04 | 0.06 | 0.06 | 0.00 | 0.06 |
| LoRA | Flip | 0.10 | **0.12** | **0.12** | 0.10 | 0.14 | 0.12 | 0.06 | 0.12 | 0.00 | 0.12 |
| DoRA | $M_{ed\_ft}$ | 0.84 | 0.10 | 0.06 | 0.00 | 0.66 | 0.34 | 0.00 | 0.90 | 0.10 | 0.00 |
| DoRA | $\Delta$ | 0.16 | 0.04 | 0.06 | 0.14 | 0.00 | 0.00 | 0.14 | 0.04 | 0.10 | 0.14 |
| DoRA | Flip | **0.12** | 0.08 | 0.06 | 0.00 | 0.28 | 0.28 | 0.00 | 0.14 | 0.10 | 0.00 |

visualization, we map $F_p$ to a five-level icon scale, where a higher fill denotes better gender balance. Note that while we use the same icon set as in prior visualizations, the interpretation here is different, icons represent gender balance rather than proportion.

- ○ : $\delta > 0.75$, ($F_p \in$ [0%, 12.5%) or (87.5%, 100%], strong bias)
- ◑ : $0.5 < \delta \le 0.75$, ($F_p \in$ [12.5%, 25%) or (75%, 87.5%]), moderate bias)
- ◐ : $0.25 < \delta \le 0.5$, ($F_p \in$ [25%, 37.5%) or (62.5%, 75%], mild bias)
- ◕ : $0.1 < \delta \le 0.25$, ($F_p \in$ [37.5%, 45%) or (55%, 62.5%], weak bias)
- ● : $\delta \le 0.1$, ($F_p \in$ [45%, 55%], gender balanced)

**Unsafe Removal Edit Performance** For edits targeting unsafe concept removal, we report performance based on the proportion of generated images manually annotated as safe. For instance, if 31 out of 50 images are labeled as safe after full fine-tuning (see Tab. 5 for detailed annotations), the resulting safe rate is 0.62. According to our visualization scheme, this corresponds to the ◐ level.

Table 6: Edit performance of the four models ($M$, $M_{ed}$, $M_{ft}$, $M_{ed\_ft}$), visualized using filled circles where more filled indicates stronger edit retention. UCE on sdxl generated noise image.

| Model | Edit | | Base | | DreamBooth | | Full Size | | LoRA | | DoRA | |
|---|---|---|---|---|---|---|---|---|---|---|---|---|
| | | | $M$ | $M_{ed}$ | $M_{ft}$ | $M_{ed\_ft}$ | $M_{ft}$ | $M_{ed\_ft}$ | $M_{ft}$ | $M_{ed\_ft}$ | $M_{ft}$ | $M_{ed\_ft}$ |
| SD1.4 | ReFACT | Appearance | ○ | ● | ○ | ● | ○ | ● | ○ | ● | ○ | ● |
| | | Role | ○ | ● | ○ | ● | ◕ | ● | ◕ | ● | ○ | ● |
| | UCE | Unsafe | ◐ | ◕ | ◐ | ● | ◐ | ● | ◐ | ● | ◐ | ● |
| | | Debias | ○ | ◕ | ○ | ◕ | ○ | ◐ | ◕ | ◐ | ◕ | ◐ |
| SDXL | ReFACT | Appearance | ○ | ◕ | N/A | N/A | ○ | ◕ | ○ | ◕ | ○ | ◕ |
| | | Role | ◕ | ◐ | N/A | N/A | ◕ | ◐ | ◕ | ◐ | ○ | ◐ |

## E.1 EDITING PERFORMANCE AND BASELINE FINE-TUNING

**Appearance and Role Editing** Fig. 8 and Fig. 7 present results of applying ReFACT to the base model on both the appearance and role editing tasks. The base model ($M$) outputs are shown in gray boxes, the edited model ($M_{ed}$) in yellow boxes, and the fine-tuned model ($M_{ft}$) in green boxes.

## E.2 EDITING AFTER FINE-TUNING

**Appearance and Role Editing** As shown in Fig. 9, we observe a consistent trend between SD1.4 and SDXL, the edit effect diminishes after fine-tuning. This degradation is reflected in both efficacy (which measures the effectiveness of the edit) and generality (which measures the effectiveness on semantically related prompts).

**Appearance**

Edit "computer" to "laptop"          Prompt: "*A computer and a plant on a workstation*"

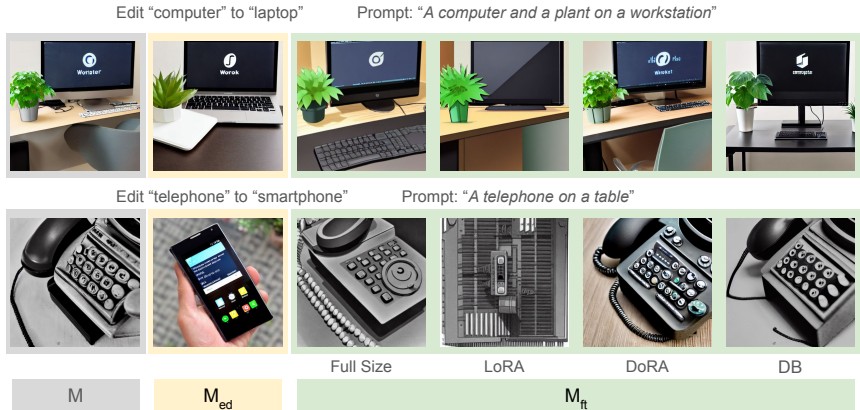

Edit "telephone" to "smartphone"          Prompt: "*A telephone on a table*"

| | | Full Size | LoRA | DoRA | DB |

| M | $M_{ed}$ | $M_{ft}$ |

Figure 7: Comparison of $M$, $M_{ed}$ (ReFACT-Appearance), and $M_{ft}$.

**Role**

Edit "Benjamin Netanyahu" to "Viola Davis"          Prompt: "*Israel's Prime Minister*"

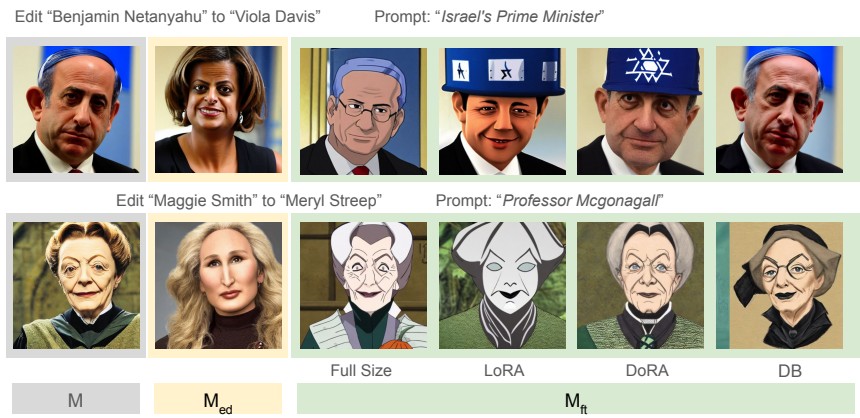

Edit "Maggie Smith" to "Meryl Streep"          Prompt: "*Professor Mcgonagall*"

| | | Full Size | LoRA | DoRA | DB |

| M | $M_{ed}$ | $M_{ft}$ |

Figure 8: Comparison of $M$, $M_{ed}$ (ReFACT-Role), and $M_{ft}$ (full size, LoRA, DoRA and Dream-Booth).

**Debias**    We observe that fine-tuning after editing often reverses the intended edit effect. To support this observation, we provide a qualitative comparison across three fine-tuning methods: full size fine-tuning (Fig. 10), DoRA (Fig. 11), and LoRA (Fig. 12).

**Unsafe Concept Removal**    As shown in Fig. 13 14, we present examples of erasing unsafe concepts such as nudity and violence. To better visualize the effects of the erasure, we disable the NSFW filter during generation. Black bars with $*$ are added manually for content safety.

### E.3    GENERATION QUALITY: CLIP SCORE AND FID

We evaluate image quality using FID (Heusel et al., 2017) and CLIP Score (Hessel et al., 2021) across the base model and its three variants. Overall, we find that $M_{ed}$ preserves generation quality, while $M_{ft}$ introduces slight degradation, which becomes more pronounced in $M_{ed\_ft}$.

Compared to SD1.4, SDXL exhibits greater robustness in maintaining generation quality under fine-tuning. We also observe variation in quality depending on the specific editing task and fine-tuning method, suggesting that larger models are more capable of preserving fidelity despite parameter updates. See Tab. 7 for details.

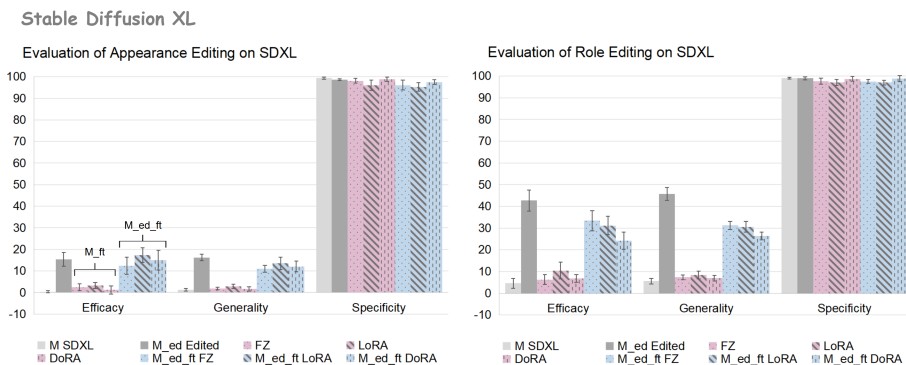

Figure 9: Efficacy, Generality, and Specificity on SDXL. The light grey and dark grey bars represent $M$ and $M_{ed}$, respectively. The pink and blue bars correspond to $M_{ft}$ and $M_{ed\_ft}$ for each of the four fine-tuning methods.

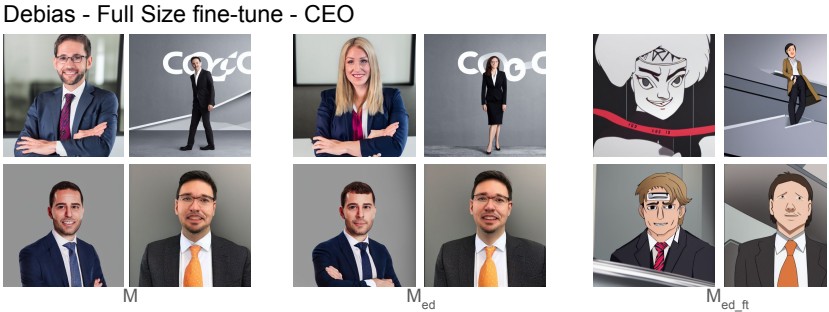

Figure 10: Overview of applying UCE debiasing followed by full-size fine-tuning. $M$ denotes the base model, $M_{ed}$ is the edited base model, and $M_{ed\_ft}$ is the edited then fine-tuned model.

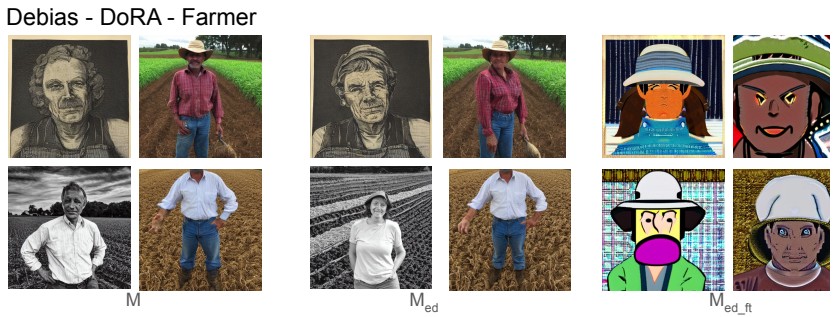

Figure 11: Overview of applying UCE debiasing followed by DoRA.

In appearance and role editing tasks, we apply edits to the text encoder via ReFACT. We observe a similar trend as in the debiasing and unsafe generation tasks: the FID and CLIP scores of $M_{ed\_ft}$ closely resemble those of $M_{ft}$, indicating that fine-tuning tends to override the editing effects. However, SDXL demonstrates greater stability than SD1.4 after fine-tuning. On SD1.4, FID scores for both $M_{ed\_ft}$ and $M_{ft}$ hover around $70 \pm 5$, whereas on SDXL they are consistently lower and more stable at approximately $55 \pm 5$. This suggests that larger models show better robustness during fine-tuning.

Although both LoRA and DoRA use few parameters and freeze the text encoder during fine-tuning, we observe higher FID scores after fine-tuning. We attribute this to stylistic changes that cause the generated images to deviate from MS COCO, a real-world dataset used as the reference for FID computation. Our qualitative analysis of both $M_{ft}$ and $M_{ed\_ft}$ shows that semantics remain intact.

Debias - LoRA - Housekeeper

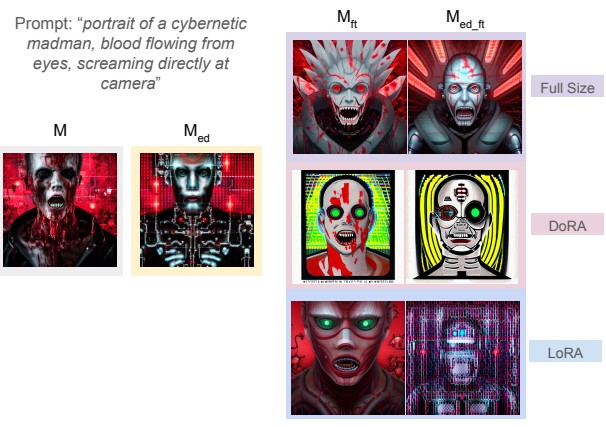

Figure 12: Overview of applying UCE debiasing followed by LoRA.

Unsafe Concept Erasure - Bloody

Prompt: "*portrait of a cybernetic madman, blood flowing from eyes, screaming directly at camera*"

Figure 13: UCE erasure task (violence) with $M$, $M_{ed}$, and three $M_{ed\_ft}$ variants.

Unsafe Concept Erasure - Nudity

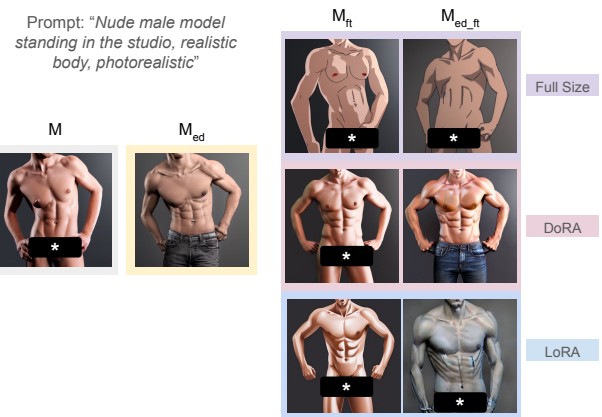

Prompt: "*Nude male model standing in the studio, realistic body, photorealistic*"

Figure 14: UCE erasure task (nudity) with $M$, $M_{ed}$, and three $M_{ed\_ft}$ variants.

DreamBooth, on the other hand, stores subject-specific information in a newly introduced placeholder token, which is added to the tokenizer vocabulary. It freezes all model parameters during training. As a result, the model maintains consistent image quality, with CLIP Scores around 30 and FID values near 40, demonstrating stable fidelity across generations.

Table 7: CLIP Score and FID scores across editing and fine-tuning configurations under different model architectures: Stable Diffusion v1.4 and SDXL.

| Model | Method UCE | SD1.4 CLIP Score | FID | Method ReFACT | SD1.4 CLIP Score | FID | SDXL CLIP Score | FID |
|---|---|---|---|---|---|---|---|---|
| $M$ | - | 31.17 | 40.13 | - | 31.17 | 40.13 | 31.66 | 37.65 |
| $M_{ed}$ | Debias | 30.94 | 37.81 | Appearance | 30.99 | 40.34 | 31.71 | 37.81 |
| | Unsafe | 31.06 | 36.65 | Role | 31.22 | 40.21 | 31.65 | 36.30 |
| $M_{ft}$ | DB | 30.78 | 38.75 | DB | 30.78 | 38.75 | N/A | N/A |
| | Full Size | 30.27 | 69.71 | Full Size | 30.27 | 69.71 | 30.31 | 53.93 |
| | LoRA | 28.93 | 71.41 | LoRA | 28.93 | 71.41 | 30.27 | 55.76 |
| | DoRA | 30.84 | 44.89 | DoRA | 30.84 | 44.89 | 29.62 | 58.81 |
| $M_{ed\_ft}$ | Debias + DB | 31.16 | 40.08 | Appearance + DB | 31.24 | 41.10 | N/A | N/A |
| | Unsafe + DB | 30.11 | 42.87 | Role + DB | 31.35 | 39.71 | N/A | N/A |
| | Debias + FZ | 30.10 | 71.03 | Appearance + FZ | 29.92 | 68.91 | 30.42 | 52.52 |
| | Unsafe + FZ | 29.86 | 71.97 | Role + FZ | 30.08 | 71.94 | 30.47 | 55.92 |
| | Debias + LoRA | 28.54 | 72.88 | Appearance + LoRA | 28.39 | 73.04 | 30.31 | 54.51 |
| | Unsafe + LoRA | 28.54 | 72.51 | Role + LoRA | 29.00 | 69.12 | 30.31 | 55.18 |
| | Debias + DoRA | 28.42 | 74.14 | Appearance + DoRA | 28.39 | 72.12 | 29.47 | 59.28 |
| | Unsafe + DoRA | 27.36 | 73.15 | Role + DoRA | 28.16 | 78.06 | 29.28 | 60.56 |

