# OpenReview forum: "On the Robustness of Model Edits under Fine-Tuning in Text-to-Image Models"
_ICLR.cc/2026/Conference — ICLR 2026 Conference Withdrawn Submission_

### Official Review · Reviewer_82ZR · 2025-10-24

**Soundness:** 2
**Presentation:** 2
**Contribution:** 1
**Rating:** 2
**Confidence:** 4

**Summary:**

This paper systematically investigates the robustness of model editing in text-to-image models after subsequent fine-tuning.  Experiments across two editing and four fine-tuning methods provide empirical evidence that model editing generally persists, albeit with important nuances.

**Strengths:**

1. Studies how edits persist after fine-tuning, providing empirical evidence across three editing tasks and different methods.

2. Offers practical guidance for selecting ''editing'' and ''fine-tuning'' methods in T2I models.

**Weaknesses:**

1. The paper presents "model editing" and "fine-tuning" as distinct terms, but both can be regarded as specific tasks of continual learning (or sequential editing) aimed at adapting model behavior. In addition, methodologically ''fine-tuning'' can itself be viewed as a specific type of ''editing'' method, such as ESD [1] for concept erasing.

2. The core finding that ''fine-tuning'' can degrade or benefit prior ''editing'' is analogous to the well-studied Backward Knowledge Transfer in continual learning [2]. Given the established literature, the paper is encouraged to position its findings better.

3. The paper could also better contextualize its findings with recent continual adaptation methods for T2I models [3,4], which specifically address task interference.

4. The experimental analysis of ''model editing'' is confined to closed-form editing methods (UCE, ReFACT), overlooking other prominent paradigms like full (or part) fine-tuning based[1], LoRA-based [5], and adapter-based [6] editing. Including more diverse editing approaches would strengthen the generalizability of the claimed findings.

[1] Erasing Concepts from Diffusion Models, ICCV 2023

[2] Beyond Not-Forgetting: Continual Learning with Backward Knowledge Transfer, NeurIPS 2022

[3] Continual Diffusion: Continual Customization of Text-to-Image Diffusion with C-LoRA, TMLR 2024

[4] How to Continually Adapt Text-to-Image Diffusion Models for Flexible Customization? NeurIPS 2024

[5] One-dimensional Adapter to Rule Them All: Concepts, Diffusion Models and Erasing Applications, CVPR 2024

[6] Receler: Reliable Concept Erasing of Text-to-Image Diffusion Models via Lightweight Eraser, ECCV 2024

**Questions:**

Please see the weaknesses.

---

### Official Review · Reviewer_7o7C · 2025-10-29

**Soundness:** 2
**Presentation:** 2
**Contribution:** 1
**Rating:** 2
**Confidence:** 4

**Summary:**

This paper investigates the robustness of model edits to subsequent fine-tuning in text-to-image diffusion models, addressing a gap in understanding whether edits persist or degrade during real-world deployment. The authors conduct experiments across two model families (Stable Diffusion and FLUX), two editing techniques (Unified Concept Editing and ReFACT), and four fine-tuning methods (full-size, DreamBooth, LoRA, and DoRA), evaluating three types of editing tasks: concept appearance/role modification, gender debiasing, and unsafe content removal. Their key findings reveal that fine-tuning slightly weakens debiasing and concept edits (with approximately 6.78% average reversal for appearance edits across fine-tuning methods), while strengthening unsafe content removal edits.

**Strengths:**

* This appears to be the first systematic study examining how model edits interact with subsequent fine-tuning in T2I models, addressing a critical gap that's highly relevant for real-world deployment scenarios.
* The counterintuitive result that fine-tuning strengthens unsafe content removal while weakening other edits is valuable for the community and has clear dual implications for both defending against malicious edits and maintaining beneficial ones.
* The paper provides concrete guidance for practitioners (e.g., use DreamBooth/LoRA to preserve edits, use full-size/DoRA to remove malicious edits).

**Weaknesses:**

- The paper is purely empirical with no mechanistic understanding of why fine-tuning affects different edit types differently. The brief speculation about polysemantic representations (Section 5) is underdeveloped and not validated. There's no formal analysis of the parameter space overlap between editing and fine-tuning operations.
* Limited experimental scale and diversity:
    * Only 50 prompts for unsafe content evaluation
    * Only 6 professions were tested for debiasing
    * Fine-tuning limited to one anime-style dataset (Naruto) - lacks diversity in domain shifts
    * No evaluation on how findings generalize to other styles (photorealistic, artistic, etc.)
* No systematic investigation of when and why edits fail after fine-tuning. The qualitative example in Figure 5 shows gradual degradation, but this isn't quantified or analyzed across multiple cases.
* The paper attributes unsafe content removal improvement to "style shift" but doesn't isolate this variable or test alternative hypotheses. The mechanism remains speculative.

**Questions:**

Q1: Can you provide any mechanistic analysis of why different edit types respond differently to fine-tuning? For instance, have you examined which specific layers or parameters are modified during editing vs. fine-tuning for each task? A simple overlap analysis (e.g., measuring parameter change correlation between editing and fine-tuning) could strengthen your claims.

Q2: You briefly mention polysemantic representations in Section 5. Have you conducted any neuron-level or feature-level analysis to validate this hypothesis? For example, could you show that neurons targeted by nudity-removal edits are indeed re-activated during Naruto-style fine-tuning, supporting your claim that "units may re-activate previously suppressed nudity"?

Q3: Can you formalize the relationship between edit robustness and fine-tuning method characteristics? For instance, why exactly does DoRA override edits more than LoRA?

Q4: Your unsafe content evaluation uses only 50 prompts from 4,703 available. Can you provide results on a larger sample (e.g., 200-500 prompts) to validate that your findings are statistically robust? Similarly, for debiasing, why were only 6 professions selected from 40 available?

---

### Official Review · Reviewer_5L19 · 2025-11-02

**Soundness:** 2
**Presentation:** 2
**Contribution:** 1
**Rating:** 2
**Confidence:** 3

**Summary:**

The paper systematically investigates whether model edits persist through subsequent fine-tuning in text-to-image diffusion models. The authors evaluate two editing methods and four fine-tuning approaches across Stable Diffusion and FLUX models on three editing tasks. The main finding is that fine-tuning slightly weakens concept and debiasing edits (approximately 6-7% reversal) but unexpectedly strengthens unsafe content removal edits. The results suggest fine-tuning can serve as both a potential defense against malicious edits and a process that may require reapplication of beneficial edits.

**Strengths:**

- The paper evaluates multiple models (SD1.4, SDXL, FLUX), editing methods (UCE, ReFACT), fine-tuning approaches (full-size, DreamBooth, LoRA, DoRA), and diverse editing tasks (appearance, role, debiasing, safety). This breadth is commendable.
- The research question is important for real-world deployment where models undergo both editing and subsequent fine-tuning. Understanding edit persistence has clear implications for AI safety and model maintenance.
- The paper correctly identifies that edit persistence has both positive implications (maintaining beneficial edits) and negative implications (difficulty removing malicious edits), though this dual perspective could be developed more thoroughly.
- The use of human annotators for unsafe content evaluation (with reported Fleiss' Kappa of 0.717) is appropriate given the limitations of automated safety classifiers, as demonstrated in their analysis.

**Weaknesses:**

- The paper treats "editing" and "fine-tuning" as conceptually distinct operations, but this distinction is not well-justified given that both involve parameter modifications. UCE modifies cross-attention projection matrices (WK and WV), while DreamBooth can use the exact same methodology (DreamBooth LoRA - open-sourced in Diffusers examples - modifies the same matrices). This inconsistency undermines the theoretical framework. The authors should either: (a) provide a principled distinction between these operations beyond their intended purpose, or (b) reframe the work as studying different types of sequential parameter updates.
- Line 160 incorrectly separates DreamBooth from a full-size fine-tuning, which is exactly how the method is described in the original paper. More critically, line 163 claims that "for LoRA and DoRA, we apply fine-tuning only to the text encoder," but these methods are standardly applied to the UNet as well as other different architectures (as evidenced by standard implementations in Diffusers). This restricted experimental setup is not justified and limits the generalizability of findings. Why were these architectural choices made, and how might results differ with arguably more standard configurations, like applying low-rank adaptations to the UNet instead?
- Equation 4 defines the discrepancy metric $\Delta$ but does not specify whether image generation uses the same initial noise across model variants or different random seeds. This is critical for reproducibility and interpretation, since if different noise is used, observed differences may partially reflect sampling variance rather than true edit robustness. Additionally, the normalization operation $\lVert \cdot \rVert$ is vaguely defined as "normalization operations based on the specific evaluation needs, for example, removing black images." This ad-hoc treatment of different evaluation scenarios should be formalized.
- The paper omits relevant work on knowledge modification and catastrophic forgetting in customized generative models. Specifically, [1] directly addresses forgetting in fine-tuned image models, which is central to understanding edit persistence. The literature on continual learning and catastrophic forgetting in LLMs [2,3] also provides relevant methodological frameworks that could strengthen the theoretical foundation.
- The caption of Figure 1 refers to editing 'safe' and 'gender' concepts, but the visual content appears to show the model learning from example datasets rather than illustrating edit effects. The relationship between the description in the text and the images is unclear.
- Section 4.6 offers minimal actionable guidance beyond noting that edits "generally persist" with "minor degradation." For practitioners deploying edited models, more specific guidance is needed: When should edits be reapplied? How can edit robustness be improved during the initial editing phase? What are the failure modes to monitor?
- The finding that unsafe content removal is strengthened by fine-tuning is potentially the most interesting result, yet it receives superficial analysis. The authors attribute this to style shift, but provide no rigorous evidence for this hypothesis. Were ablations conducted with photorealistic fine-tuning data? Are there alternative explanations? This result deserves deeper investigation as it has important implications for AI safety.


While the paper addresses a practically relevant question, it suffers from fundamental issues in problem formulation, unjustified experimental restrictions, incomplete analysis, and limited contributions.

[1] "Assessing Open-world Forgetting in Generative Image Model Customization", Laria et al., 2024
[2] "Revisiting Catastrophic Forgetting in Large Language Model Tuning", Li et al., EMNLP 2024
[3] "An Empirical Study of Catastrophic Forgetting in Large Language Models During Continual Fine-tuning", Luo et al., 2024

**Questions:**

The most pressing questions were already stated in the weaknesses section, i.e., problem formulation, restrictions, noise seeding, etc.

- All fine-tuning was done with anime-style data. How do your findings generalize to other common fine-tuning scenarios (e.g., photorealistic style adaptation, domain-specific fine-tuning on medical images, etc.)?

---

### Note · Authors · 2026-01-17

I have read and agree with the venue's withdrawal policy on behalf of myself and my co-authors.